Investigating the potential use of Sentinel-1 data for monitoring wetland water level changes in China’s Momoge National Nature Reserve

Chen Yueqing 1 2 3
Qiao Sijia 2
Zhang Guangxin zhgx@iga.ac.cn 1
Xu Y. Jun 4
Chen Liwen 1
Wu Lili wulili@iga.ac.cn 2 3
1 Northeast Institute of Geography and Agroecology, Chinese Academy of Sciences , Changchun , Jilin , China
2 School of Geographic Sciences, Xinyang Normal University , Xinyang , Henan , China
3 Key Laboratory for Synergistic Prevention of Water and Soil Environmental Pollution, Xinyang Normal University , Xinyang , Henan , China
4 School of Renewable Natural Resources, Louisiana State University Agricultural Center , Baton Rouge , America
Wilson Matthew
Electronic publication date: 2020 Feb 17
Publication date: 2020
Volume: 8
Electronic Location ID: e8616
Received 2019 Nov 21; Accepted 2020 Jan 21
Copyright: ©2020 Chen et al.
Copyright year: 2020
Copyright holder: Chen et al.
License: This is an open access article distributed under the terms of the Creative Commons Attribution License, which permits unrestricted use, distribution, reproduction and adaptation in any medium and for any purpose provided that it is properly attributed. For attribution, the original author(s), title, publication source (PeerJ) and either DOI or URL of the article must be cited.
License URL: https://creativecommons.org/licenses/by/4.0/

Keywords: Backscatter, Interferometric coherence, Wetland, Water level, Interferometric synthetic aperture radar (InSAR), Sentinel-1, Momoge National Nature Reserve

Funding: National Key R & D Program of China 2017YFC0406003 National Natural Science Foundation of China 41701395 41877160 41671476 Featured Institute Project 4 Northeast Institute of Geography and Agroecology, Chinese Academy of Sciences IGA-135-05 Nanhu Scholars Program for Young Scholars of XYNU U.S. Department of Agriculture Hatch Fund project LAB94459 This research was funded by the National Key R & D Program of China (grant number 2017YFC0406003) the National Natural Science Foundation of China (grant number 41701395, 41877160 and 41671476) the Featured Institute Project 4, the Northeast Institute of Geography and Agroecology, Chinese Academy of Sciences (grant number IGA-135-05) and the Nanhu Scholars Program for Young Scholars of XYNU. During the preparation of the article, Yijun Xu received funding support from a U.S. Department of Agriculture Hatch Fund project (project number: LAB94459). The funders had no role in study design, data collection and analysis, decision to publish, or preparation of the manuscript.

==============================
Background

Interferometric Synthetic Aperture Radar (InSAR) has become a promising technique for monitoring wetland water levels. However, its capability in monitoring wetland water level changes with Sentine-1 data has not yet been thoroughly investigated.

Methods

In this study, we produced a multitemporal Sentinel-1 C-band VV-polarized SAR backscatter images and generated a total of 28 interferometric coherence maps for marsh wetlands of China’s Momoge National Nature Reserve to investigate the interferometric coherence level of Sentinel-1 C-VV data as a function of perpendicular and temporal baseline, water depth, and SAR backscattering intensity. We also selected six interferogram pairs acquired within 24 days for quantitative analysis of the accuracy of water level changes monitored by Sentinel-1 InSAR. The accuracy of water level changes determined through the Sentinel-1 InSAR technique was calibrated by the values of six field water level loggers.

Results

Our study showed that (1) the coherence was mainly dependent on the temporal baseline and was little affected by the perpendicular baseline for Sentinel-1 C-VV data in marsh wetlands; (2) in the early stage of a growing season, a clear negative correlation was found between Sentinel-1 coherence and water depth; (3) there was an almost linear negative correlation between Sentinel-1 C-VV coherence and backscatter for the marsh wetlands; (4) once the coherence exceeds a threshold of 0.3, the stage during the growing season, rather than the coherence, appeared to be the primary factor determining the quality of the interferogram for the marsh wetlands, even though the quality of the interferogram largely depends on the coherence; (5) the results of water level changes from InSAR processing show no agreement with in-situ measurements during most growth stages. Based on the findings, we can conclude that although the interferometric coherence of the Sentinel-1 C-VV data is high enough, the data is generally unsuitable for monitoring water level changes in marsh wetlands of China’s Momoge National Nature Reserve.

Introduction

Covering between 1–2% of the Earth’s surface, wetlands are among the most productive ecological systems and perform important eco-hydrological functions such as food supply, water storage and purification, flood control, climate change mitigation, as well as provide a desirable habitat for wildlife (Millenium Ecosystem Assessment, 2005; Costanza et al., 1997; Mitsch & Gosselink, 2015). Unfortunately, according to the latest IPBES assessment report (Scholes et al., 2018), wetlands are particularly degraded, with 87% lost globally in the last 300 years, and 54% since 1900. It is widely recognized that wetlands play an important role in the hydrological cycle, and vice versa, the hydrological conditions have a crucial impact on the development and degradation of wetlands (Cole, Brooks & Wardrop, 1997; Hunt, Walker & Krabbenhoft, 1999; Mitsch & Gosselink, 2015). Therefore, the conservation and restoration of wetlands requires more specific information about their hydrological conditions. Typically, hydrological monitoring of wetlands is carried out through stage (water level) stations providing good temporal resolution over a finite number of observation points (Hong et al., 2010a). However, these in-situ measurements have limited capability to detect spatial patterns, as gauge stations are usually located several or even tens of kilometers from each another (Hong et al., 2010a; Wdowinski & Hong, 2015).

Wetland InSAR technique can be an excellent complementary tool for in-situ ground observations to better understand and monitor a wide area with high spatial resolution (Hong & Wdowinski, 2017). Since the first time Alsdorf et al. (2000) and Alsdorf, Smith & Melack (2001) mapped a spatial detailed image of centimeter-scale variations in the Amazon floodplain water level response to changing river discharge through InSAR, innovative applications of InSAR to monitoring hydrologic changes in wetlands have also been successful in different regions of the world (Kim et al., 2014), including but not limited to the Everglades (Hong et al., 2010a; Kim et al., 2014; Liao & Wdowinski, 2018; Wdowinski et al., 2004; Wdowinski et al., 2008), the Louisiana wetlands (Kim et al., 2009; Kwoun & Lu, 2009; Lu et al., 2005), the Amazon floodplain (Cao et al., 2018), the Sian Ka’an in Yucatan (Gondwe et al., 2010), the Yellow River Delta (Xie et al., 2013; Xie et al., 2015; Yuan et al., 2016), the Liaohe River (Zhang et al., 2016), the Great Dismal Swamp (Kim et al., 2017), the Ciénaga Grande de Santa Marta (Jaramillo et al., 2018), the Yukon Flats Basin (Pitcher et al., 2019) and, most recently, the Peace-Athabasca Delta (Siles et al., 2020). Today, wetland InSAR technique has evolved from monitoring relative water level changes to monitoring absolute water level time series. However, using the InSAR technique for monitoring wetland water level is still a relatively new research field that has not yet been fully exploited (Mohammadimanesh et al., 2018b), especially for many developing and undeveloped countries.

Radar signal backscattering mechanisms can be simplified into four major categories: double bounce scattering, surface scattering, volume scattering, and specular scattering (Kwoun & Lu, 2009). The wetland InSAR technique works where vegetation emerges above the water surface owing to the double bounce effect, in which the radar signal is backscattered twice from the water surface and vegetation (Richards, Woodgate & Skidmore, 1987). Three main sources of losing interferometric coherence, i.e., geometric, volumetric and temporal decorrelations, are integrated and determine the portion of the SAR signal that is available to produce double bounce backscattering over wetlands (Lu & Kwoun, 2008). Furthermore, different wetland classes such as marsh, swamp, bog, fen, and shallow water have different backscattering behavior depending on SAR satellites wavelength, polarization, incidence angle, spatial resolution, environmental variables, and wetland phenology (Mohammadimanesh et al., 2018b). To date, a number of studies have discussed the potential of using different SAR data sources such as ERS-1/2 (Lu et al., 2005; Lu & Kwoun, 2008), JERS-1 (Wdowinski et al., 2004; Wdowinski et al., 2008), RADARSAT-1/2 (Brisco et al., 2017; Gondwe et al., 2010; Kim et al., 2017; Kim et al., 2009; Lu & Kwoun, 2008; Mohammadimanesh et al., 2018a; Siles et al., 2020), ENVISAT (Wdowinski et al., 2006), ALOS PALSAR-1/2 (Cao et al., 2018; Jaramillo et al., 2018; Kim et al., 2017; Kim et al., 2014; Kim et al., 2009; Mohammadimanesh et al., 2018a; Palomino-Ángel et al., 2019; Yuan, Lee & Jung, 2017) and TerraSAR-X (Hong, Wdowinski & Kim, 2010b; Mohammadimanesh et al., 2017) to detect water level changes in different types of wetlands. However, most SAR satellites that provide data for previous wetland InSAR studies have a relatively short life span and have been out of operation for years or even two decades. In addition, most of these SAR satellites have long repeat observation cycle (11–46 days), limited swath (15–80 km), and limited accessibility (often requires payment).

The Sentinel-1 satellite launched in 2014 is the second latest SAR mission operating at C-band. It provides SAR datasets with a short repeat cycle of 6–12 days and a wide coverage area (250 km) (Liao & Wdowinski, 2018), while the perpendicular baseline is well controlled (Torres et al., 2012). The cost-free availability of Sentinel-1 data is also an encouraging factor to investigate the potential use of such data for wetland InSAR applications (Mohammadimanesh et al., 2018b). Despite these benefits, there has been little quantitative analysis of using Sentinel-1 data to monitor wetland water level changes. Kundu et al. (2017) applied the Sentinel-1 data to estimate water level changes during the floods in the Lember Basin in October 2016. Hong & Wdowinski (2017) and Liao & Wdowinski (2018) reported that the interferograms using Sentinel-1 data show distinct fringe patterns related to water level changes over the Everglades. Zhang et al. (2018) used the Sentinel-1 data to extend the precise measurement of a single water gauge to a wide area of 29 km2 in the Palmdale in South Florida. Alexakis, Stavroulaki & Tsanis (2019) discovered that in Agia Lake and Kournas Lake, low vegetation is the most critical parameter that causing volume scattering, leading to low interferometric coherence, and is therefore the limiting factor in estimating water level changes using Sentinel-1 data. Chen et al. (2019) found that in the marsh wetlands of the Great Lakes, the interferometric coherence derived from the Sentinel-1 data is highly correlated with its temporal baseline (i.e., interferogram’s time span). However, in wetlands, the relation between the variation of Sentinel-1 interferometric coherence and backscattering intensity is largely unknown. In addition, given that the use of sentinel-1 data for monitoring wetland water level changes is limited to specific pilot sites, the potential use of such data in wetlands with different vegetation characteristics and environmental variables has not been extensively studied.

This study aims to systematically investigate the potential use of Sentinel-1 C-VV data to detect water level changes in marsh wetlands. We used China’s Momoge National Nature Reserve as a case study here because it is widely representative of relatively shallow marshes. Also, this reserve is recognized by the Ramsar Convention on Wetlands of International Importance, where each year over 90% of the world’s Siberian Cranes stage on migration in the large shallow water area. Specifically, the study has the following objectives: (1) to analyze the response of Sentinel-1 interferometric coherence to the perpendicular and temporal baseline, and seasonal fluctuation of water depth; (2) to determine the relation between Sentinel-1 interferometric coherence and backscattering intensity; and (3) to assess the influence of the coherence level, the stage of the growing season, and the water depth on the accuracy of monitoring water level changes in marsh wetlands using the Sentinel-1 InSAR technique.

Materials and Methods

Study area

The Momoge National Nature Reserve (MNNR; 45°42′25″–46°18′00″N, 123°27′00″–124°04′33.7″E) is located in the West Jilin Province, Northeast China, covering an area of 1,440 km2 (Fig. 1). The reserve has wetland types that are typical to the biogeographic region, such as low plain marshes and shallow lakes. Two large rivers flow through the area, the Nenjiang and Tao’er Rivers, and there are several man-made ditches connecting the water bodies in the reserve. These habitats provide important refuge for a variety of fish, bird, and other wildlife populations. In spring 2012, 97% of the world’s population of the critically endangered Siberian Crane Leucogeranus were recorded at the site, and over 100,000 water birds were recorded in each year between 2010 and 2012 (https://rsis.ramsar.org/ris/2188). In the light of the above, the MNNR was included on the List of Wetlands of International Importance (Ramsar sites) in 2013. The Baihe Lake formerly known as Etou Lake, which is located in the middle of the MNNR, is the main stopover site for Siberian Cranes (Wang et al., 2013). The region has a temperate continental monsoon climate with an annual precipitation of 389.4 mm, an annual temperature of 4.2 °C, and an annual evaporation (E601) rate up to 1,000 mm. Precipitation is concentrated in the summer, making more than 90% of the annual total precipitation during the period from May to October. Fig. 2 depicts the total precipitation (mm) in the study area for each month in 2016.

Figure 1 Location map of the study area showing the extent and DEM of the Momoge National Nature Reserve (MNNR) in northeast China.

Six experimental Odyssey® Capacitance water level loggers were installed in the marsh wetlands of the Baihe Lake in the study area. Data source: NASA Earth data.

Figure 2 Monthly precipitation in the study area in 2016.

Data were collected from the China Meteorological Data Service Center (http://data.cma.cn).

Field data collection

Six experimental Odyssey® Capacitance water level loggers were installed in the marsh wetlands of the Baihe Lake to automatically monitor the water level in 2016. The six automated water level monitoring sites are located at (45°54′25″N, 123°41′08″E), (45°55′59″N, 123°37′14″E), (45°53′43″N, 123°37′22″E), (45°54′08″N, 123°44′13″E), (45°53′28″N, 123°43′19″E) and (45°52′56″N, 123°44′42″E), respectively. The water level loggers at sites 1, 2, 4 and 5 were set to take hourly records for the entire study period from May 27, 2016 to October 24, 2016, while the logger at site 3 was set to take records at the same frequency from June 05, 2016 to October 24, 2016. The acquisition time (22:00) of the water level/depth data we used was closest to the transit time (∼22:00) of the Sentinel-1 radar satellite. Records from the six field water level loggers were used to calibrate the water level changes obtained through the Sentinel-1 InSAR technique.

Satellite data collection

Optical satellite image

The Landsat 8 OLI Land Surface Reflectance Level-2 data product at a 30-meter spatial resolution acquired on October 6, 2016 were gathered from the USGS web portal (https://earthexplorer.usgs.gov/) and were used to generate the land use and land cover (LULC) map for the study area.

SAR satellite image

A total number of eight VV-polarized Sentinel-1A/B C-band Level-1 images in Interferometric Wide Swath Mode (IW) Single Look Complex (SLC) format acquired on May 27, June 08, July 26, August 07, August 19, September 12, September 30 and October 24 were downloaded from the ESA web portal (https://scihub.copernicus.eu/). These images were used to generate backscatter, coherence, interferograms and water level change maps. The Sentinel-1 synthetic aperture radar (SAR) instrument has a spatial resolution of 5 m × 20 m. All Sentinel-1 SAR images completely covered the study area.

DEM

A digital elevation model (DEM) with a spatial resolution of 12.5 m derived from ALOS PALSAR-1 imagery acquired on November 12, 2006 was obtained from the Alaska Satellite Facility web portal (https://www.asf.alaska.edu/). The DEM was employed to geocode the SAR images as well as flatten the interferograms.

Methodology

Steps of the methodology

The methods mainly include supervised maximum likelihood classification (MLC), InSAR technique and least square fit analysis. Fig. 3 describes the detailed steps of the methodology.

LULC classification based on maximum likelihood classifier

LULC classification is the stage of image analysis in which the multivariate quantitative measurement associated with each pixel is translated into a label from a pre-defined land use category (Deilmai, Ahmad & Zabihi, 2014). The main steps of LULC classification may include determination of a suitable classification system, image pre-processing, selection of training samples, feature extraction, selection of suitable classification approaches, post-classification processing and accuracy assessment (Deilmai, Ahmad & Zabihi, 2014).

Figure 3 Flowchart of the image processing and data analysis in this study.

Before classification, eight land cover types were defined for the study area based on our knowledge of the area, including Marsh, Water, Ditch, Dam, Cropland, Forest, Residential and Saline Soil. The ENVI software (version 5.5.2, Harris Geospatial Solutions, Inc., Broomfield, CO, USA) was used for image processing and analysis. During the pre-processing stage, the seven single-band image files were stacked into one multi-band image file. Then, the image was cropped to the study area size. After that, band combinations 432, 543, 564 and 654 were used flexibly to make composite images. Due to the lack of ground truth samples, training samples were interactively created on the composite images using the “Region of Interest (ROI)” tools provided by ENVI Software with the assistance of the Google Earth historical images acquired in 2016 (Jia et al., 2014). The spatial resolution of the Google Earth historical images ranged from 0.21 m to13.30 m. The sizes of the ROIs varied depending on the land cover features. In addition, the ROI separability (i.e., Jeffries-Matusita and Transformed Divergence) was computed and the ROIs were re-selected until the value of Jeffries-Matusita and Transformed Divergence was greater than 1.85. Table 1 summarizes the characteristics of the final ROIs.

Table 1 Number of ROIs and pixels in each class type used for training the MLC classifier.

	Marsh	Water	Ditch	Dam	Cropland	Forest	Residential	Saline soil	
Number of ROIs	198	197	48	52	187	156	82	138	
Number of pixels	5,781	6,672	553	589	3,274	4,681	2,780	3,810	

MLC is especially suitable for the classification of moderate resolution remote sensing images (e.g., Landsat 8 OLI images with a spatial resolution of 30 m in this study) and has a high overall accuracy (mostly over 80%) (Deilmai, Ahmad & Zabihi, 2014). For this reason, we used MLC rule for the spectral classification of the Landsat 8 OLI images. After classification, the “Edit Classification Image” tools were used for post-classification processing refers to the process of removing the noise and improving the quality of the classified output. After post-classification processing, the accuracy of the classified images was assessed using reference data (i.e., the previously mentioned Google Earth historical images of year 2016). A total of 240 points were selected from the Google Earth historical images generating via a stratified random sampling method. Precision (user’s accuracy), recall (producer’s accuracy), F1, overall accuracy, and the Kappa statistics were derived from the confusion matrix to find the accuracy and reliability of the maps produced (Islam et al., 2018; Tharwat, 2018).

Backscatter generation

The SARscape (version 5.5.2) which is a modular set of ENVI software was applied to generate backscatter for the multitemporal SAR images. The processing consisted of several steps, including reading the Sentinel-1 SLC time series images, generating the ‘four range and one azimuth’ multilook intensity images to enhance the radiometric resolution of the radar signal and the signal/noise ratio, co-registration, filtering through a Gaussian Gamma MAP filter with a kernel of 5 × 5 pixels to decrease the speckle, and using the Range-Doppler approach for geocoding. After that, the obtained intensity values were converted into normalized backscattering coefficient (σ0) values in decibel.

After producing SAR backscatter images, an analysis of σ0 variation was carried out for the marsh wetlands. We use a multitemporal backscattering response for Sentinel-1 C-VV to identify stable/unstable areas in time. Using all available imagery, we calculated backscatter standard deviation values (SDσ0) in dB. We selected threshold value of 2 dB (Kim et al., 2013) for sentinel-1 to determine stable/unstable backscatter areas. Pixels with SDσ0 lower than 2 dB were considered as stable scatters areas, while with SDσ0 higher than 2 dB were considered as unstable scatters areas. The marsh wetlands were further divided into two categories: stable areas of the marsh wetlands and unstable areas of the marsh wetlands accordingly. In the next section, some analyses were carried out in stable areas of the marsh wetlands to minimize seasonal effects associated with the changes of seasonal water level and vegetation, some analyses were carried out in unstable areas of the marsh wetlands to investigate the seasonal effects, and the rest analyses were carried out in the whole marsh wetlands. ArcGIS software (version 10.7, ESRI Inc., Redlands, CA, USA) was applied for the above processing.

InSAR processing

The InSAR Processing includes coherence generation, interferogram generation, and water level changes map generation.

Interferometric coherence calculation is a well-known method to examine the quality of the interferograms and represents the degree of similarity of the same pixel in the time span between two SAR acquisitions (i.e., the so-called master and slave SAR images) (Brisco et al., 2015; Guarnieri & Prati, 1997; Mohammadimanesh et al., 2018a). Coherence is calculated by cross-correlation of the master and slave SAR images over a small window of pixels (Ferretti et al., 2007): (1) γ=s1s2∗s1s1∗s2s2∗

where S 1 and S2 denote the complex pixel values of backscattering coefficient, ∗ refers to the complex conjugate, and pixel values within <> denote their spatial averaging over a selected window size. γ ranges from 0 (low) to 1 (high); γ is equal to 1 when the two images are exactly the same, whereas γ is equal to 1 when the two images are do not correspond.

In this study, all interferometric processing was carried out using the SARscape module. We started with a baseline estimation to obtain information about both the perpendicular and the temporal baseline values in a multi-temporal SAR acquisitions series. The values for all possible interferometric pair combinations were calculated. After that, we generated “four range and one azimuth” multilook images to enhance the radiometric resolution of the radar signal and the signal/noise ratio, increase the interferometric coherence, and speed up the computing process. A digital elevation model with a spatial resolution of 12.5 m was employed to flatten the interferograms through the removal of the constant phase due to the acquisition geometry. Interferograms were filtered for visual inspection and for identification of fringe patterns through a Goldstein filter (Goldstein & Werner, 1998) with a 5 × 5 size filtering moving window for the coherence estimation. Given that the interferometric phase could not be maintained in herbaceous wetlands when the interferometric coherence was lower than 0.2 (Kim et al., 2013), we subsequently selected threshold of coherence as 0.2 for unwrapping this phase change with the minimum cost flow unwrapping algorithm and using the Range-Doppler approach for geocoding. It is worth noting that, coherence lower than the unwrapping threshold value (0.2) had not transformed from phase into water level change. At last, we converted the unwrapped phase change to water level change (Δh) employing the following equation: (2) Δh=−λΔϕ4πcosθ+n

In this equation, λ and θ are the Sentinel-1 C-VV SAR wavelength (0.055 m) and incidence angle (39.1°) respectively, Δϕ forms the extracted phase change along the line-of-sight of the satellite in each pixel of the entire study area (in terms of 2π), and n is the noise mainly caused by the above mentioned decorrelation effects. Each color cycle corresponded to approximately 0.035 m of water level changes in vertical direction, that is Δh.

Least square fit analysis

We used a least square fit analysis to evaluate the relationship between Sentinel-1 interferometric coherence and backscatter variation. The relation was described by the following equation: (3) y=a+bx

where a and b denote the offset and slope, respectively. Only interferometric coherence pairs with the smallest temporal baselines were used and the mean backscatter images were generated by averaging two SAR images, which produced the corresponding coherence images (Mohammadimanesh et al., 2018a).

In order to assess the accuracy of the InSAR-observed water level changes, we used a least square fit between InSAR and logger observations with the line passed through as many points as possible. Because the InSAR and logger observations differ by unknown offset, we addressed the problem by assuming a slope of 1 between two observations (Hong et al., 2010a; Wdowinski et al., 2008). The relation was described by the following equation: (4) y=x+offset

Origin software (version 2018C, Origin Lab Corporation, Northampton, MA, USA) was applied for the above processing.

Results

Land use land cover analysis

Much of the Momoge National Nature Reserve in 2016 was covered by cropland (Fig. 4), which contributed 37.0% of the total area. With 28.9%, marsh wetlands in the reserve were the second largest cover type. The other cover types included 18.9% for water, 7.6% for bare saline soil, 4.9% for forest, and 1.5% for residential area. Furthermore, in addition to water surface area of the two natural rivers, man-made ditches accounted for 0.64% of the reserve area; the dam class that obstructed hydrological connectivity accounted for 0.59%. Tables 2 and 3 summarize the results of the confusion matrix and the classification metrics, respectively. As seen, the classification accuracy was high and reliable.

Figure 4 Land use and land cover (LULC) distribution in the Momoge National Nature Reserve, Northeast China.

The map was created with the supervised classification based on satellite image acquired in 2016. Data source: Earth Explorer USGS data.

Table 2 Summary of confusion matrix.

		Reference Data	
		Marsh	Water	Ditch	Dam	Cropland	Forest	Residential	Saline Soil	
Classified data	Marsh	23	1	0	0	1	1	0	0	
Water	3	33	0	0	1	0	0	3	
Ditch	0	0	20	0	0	0	0	1	
Dam	0	0	2	18	0	0	0	0	
Cropland	0	1	0	0	36	2	0	0	
Forest	0	0	0	0	1	21	0	0	
Residential	0	0	0	0	0	1	44	0	
	Saline Soil	2	1	0	0	1	0	1	22	

Table 3 Summary of accuracy (%) and Kappa statistics of MLC map.

Classification metric	Marsh	Water	Ditch	Dam	Cropland	Forest	Residential	Saline soil	
Precision (User’s accuracy)	88.5	82.5	95.2	90.0	92.3	95.5	97.8	81.5	
Recall (Producer’s accuracy)	82.1	91.7	90.9	100.0	90.0	84.0	97.8	84.6	
F1	85.2	86.8	93.0	94.7	91.1	89.4	97.8	83.0	
Overall accuracy	90.4	
Kappa statistics	88.9	

Backscatter analysis

Figure 5 depicts stable and unstable backscatter areas of the marsh wetlands. The stable and unstable areas of the marsh wetlands cover 249.72 km2 and 172.49 km2, respectively. In fact, fieldwork evidence suggests that the vegetation types of the stable areas were mainly Carex tato and Calamagrostis angustifolia communities that widely distributed along the west bank of Nenjiang River, while the unstable areas were mainly Phragmites australis and Scripus triqueter communities.

Figure 5 Marsh wetlands distribution in the Momoge National Nature Reserve, Northeast China, classified with Sentinel-1multitemporal SAR images from May 2016 to October 2016.

Pixels with backscattering standard deviation value lower than 2 dB were considered as stable scatters, while with backscattering standard deviation value higher than 2 dB were considered as unstable scatters. Data source: Copernicus.

Coherence analysis

Based on analysis of 28 coherence maps, we found that the perpendicular (geometrical) baseline varies within the range between 4.85 m and 146.26 m (Fig. 6), far less than the critical perpendicular baseline which was about 5188 m. The well controlled perpendicular baseline makes the Sentinel-1 C-VV interferometric coherence independent of the perpendicular baseline (Fig. 6). We also found that the temporal baseline ranges from 12 days to 150 days (Fig. 6). Besides, the coherence was high (higher than 0.3 in most cases) over the relatively short temporal baseline (i.e., <24 days) and low (lower than 0.3 in most cases) over the relatively long temporal baseline (i.e., >24 days) (Fig. 6). The results indicate that the Sentinel-1 C-VV interferometric coherence is strongly dependent on the temporal baseline.

Figure 6 Temporal vs. perpendicular baselines in the Sentinel-1 coherence analysis for stable areas of the marsh wetlands in the Momoge National Nature Reserve, Northeast China.

Dot sizes and colors are proportional to different coherence values.

Comparing Figs. 7 and 8 shows the response of Sentinel-1 C-VV interferometric coherence to the changes of seasonal water depth. As seen in Fig. 7, in general, the mean coherence value decreased dramatically from the interferometric pair of May 27, 2016 and June 08, 2016 to the interferometric pair of August 19, 2016 and September 12, 2016 and reached its minimum value (0.31), while increased greatly from the interferometric pair of August 19, 2016 and September 12, 2016 to the interferometric pair of September 30, 2016 and October 24, 2016 and reached its maximum value (0.50). As seen in Fig. 8, most of the average water depth on the acquired dates of the interferometric pairs were lower than 1 m, except for all of the interferometric pairs at site 1, the interferometric pair of August 19, 2016 and September 12, 2016 at sites 5 and 6, as well as the interferometric pair of September 12, 2016 and September 30, 2016 and the interferometric pair of September 30, 2016 and October 24, 2016 at sites 3, 5 and 6, which were higher than 1 m. It is worth noting that, in general, the mean water depth presented an opposite variation characteristic compared with the mean coherence value from the interferometric pair of May 27, 2016 and June 08, 2016 to the interferometric pair of August 19, 2016 and September 12, 2016 (see Figs. 7 and 8), indicate that the coherence is strongly influenced by the changes of seasonal water depth. However, the mean coherence value presented an similar variation characteristics compared with the mean water depth between the interferometric pair of August 19, 2016 and September 12, 2016 and the interferometric pair of September 12, 2016 and September 30, 2016, this is because the mean water depth is not the only factor affecting the coherence, vegetation canopy could also influence the coherence and maybe the dominant factor from late August to the end of September in the study area.

Figure 7 Sentinel-1 coherence analysis results for unstable areas of the marsh wetlands.

The x-axis shows the six interferometric pairs with a short temporal baseline as no more than 24 days.

Figure 8 Mean water depths recorded by field water loggers at six monitoring sites in the marsh wetlands of the Baihe Lake, Northeast China.

The x-axis shows the acquisition dates of the six interferometric pairs with a short temporal baseline as no more than 24 days.

Figure 9 analysis indicates a negative linear relationship between Sentinel-1 C-VV interferometric coherence and backscattering, suggesting that a high coherence is a good indicator of a low backscattering response for the marsh wetlands. It is worth noting that, although the temporal baseline was within 24 days, the mean coherence was lower than 0.55. Besides, the mean backscatter coefficient shows a relatively narrow range of variation between −11.74 and −10.58.

Figure 9 Relation between mean interferometric coherence and mean SAR backscatter for stable areas of the marsh wetlands in the Momoge National Nature Reserve, Northeast China with the Sentinel-1 data.

Interferogram analysis

Interferograms of phase changes are listed in Fig. 10. Comparing Figs. 7 and 10, it can be found that although the mean coherence of the interferometric pairs was high, most of the interferograms did not exhibit distinct fringes except for the interferometric pair of July 26, 2016 and August 07, 2016 (see Fig. 10B). In addition, the phase changes did not complete a full phase change (i.e., <2π) in most areas for the interferometric pair of July 26, 2016 and August 07, 2016, indicate that the water level change was less than 0.035 m in vertical direction in most areas.

Figure 10 Interferograms of phase changes in the Momoge National Nature Reserve, Northeast China. Each interferometric pair has a short temporal baseline as no more than 24 days.

(A) Interferogram of 20160527_20160608. (B) Interferogram of 20160726_20160807. (C) Interferogram of 20160807_20160819. (D) Interferogram of 20160819_20160912. (E) Interferogram of 20160912_20160930. (F) Interferogram of 20160930_20161024. Data source: Copernicus.

Water level change analysis

The calibration was conducted between the InSAR data and the logger data (Fig. 11). The calibration plots show poor agreement between InSAR and logger data for all interferometric pairs, no matter in relatively shallow flooding marsh wetlands (water depth lower than 1 m) (Fig. 11A), or in relatively deep flooding marsh wetlands (water depth higher than 1 m) (Fig. 11B). When the calibration plot shows a good agreement between InSAR and in-situ data for the interferometric pair of July 26, 2016 and August 07, 2016, only four observations are available (Fig. 11C).

Figure 11 Calibration plots for estimating the offsets between InSAR observations and logger observations for (A) all interferometric pairs in relatively shallow flooding marsh wetlands (water depth lower than 1 m); (B) all interferometric pairs in relatively deep flooding marsh wetlands (water depth higher than 1 m); and (C) the interferometric pair of July 26, 2016 and August 07, 2016 in relatively shallow flooding marsh wetlands (water depth lower than 1 m).

The symbol “+” marks outliers that are omitted from the calibration offset calculations.

Discussion

As expected, no dependency of the interferometric coherence was found with the perpendicular baseline for Sentinel-1 C-VV data. However, interferometric coherence appeared to be strongly dependent on the temporal baseline (Fig. 6), which agrees with that reported by Chen et al. (2019). It is also worth noting that a temporal baseline of no more than 24 days is required to maintain a coherence of greater than 0.3 for Sentinel-1 C-VV data, while the coherence would be lower than 0.3 if the temporal baseline exceeds 24 days in most cases (Fig. 6). Currently, the Sentinel-1 A and B constellation can deliver a six-day repeat cycle in Europe. Besides, a new generation of SAR satellites, the C-band RADARSAT Constellation Mission (RSM) with a resolution of 1 m × 3 m and a repeat cycle of 4 days, was launched by the Canadian Space Agency in June 2019. Thus, more work is needed to investigate whether a better coherence could be obtained in the even shorter temporal baselines (i.e., six days and four days interval) for C-band SAR data.

The negative correlation between Sentinel-1 coherence and water depth during the early stage of growing season (Figs. 7 and 8) indicates the importance of considering seasonality in wetland InSAR analysis. This is because the deeper a water body is and the larger the water surface area is, the lower the double bounce scattering will be during the early growth stage of emerging vegetation when plants have a less developed canopy. However, during the late stage of the growing season (from middle September to late October), the Sentinel-1 coherence has no close relationship with water depth (Figs. 7 and 8). This may probably be due to the developed vegetation canopy, which led to high double bounce scattering, making it the main factor determining interferometric coherence. This finding is different than that reported by Alexakis, Stavroulaki & Tsanis (2019) who found that coherence values decreased with increasing NDVI in Agia Lake in Greece during a summer period.

A nearly linear negative correlation was found between Sentinel-1 C-VV interferometric coherence and backscatter for the marsh wetlands in China’s Momoge National Nature Reserve (Fig. 9), indicating that a high coherence is a good indicator of a low Sentinel-1 backscattering response for the marsh wetlands. Our finding differs from previous studies (Kim et al., 2013; Mohammadimanesh et al., 2018a) which found that there is no linear correlation between interferometric coherence and backscattering for the marsh wetlands when using JERS-1, RADARSAT-1/2, ERS-1/2, ALOS PALSAR-1 and TerraSAR-X data. In addition, the range of the Sentinel-1 C-VV backscatter was relatively narrow for the stable areas of the marsh wetlands (i.e., mainly Carex tato and Calamagrostis angustifolia communities) in the MNNR (Fig. 9) as compared to that of reed marshes in Balikdami wetland in Turkey (Kaplan & Avdan, 2018). More work is needed in the future to distinguish whether the main backscattering mechanism in the MNNR is double bounce scattering or volume scattering.

For the first time, we have found that once the coherence exceeds a certain threshold (0.3 in this study), the stage during the growing season, rather than the coherence, is the primary factor determining the quality of the interferogram for the marsh wetlands, even though the quality of the interferogram largely depends on the coherence. Comparing Figs. 10 and 12, it can be found that the interferometric pair of July 26, 2016 and August 07, 2016 exhibits the best quality of the interferogram but the third highest coherence. However, most of the interferograms in the study area did not exhibit distinct fringes, which is contrary to that reported by Hong & Wdowinski (2017) and Liao & Wdowinski (2018) for the Florida Everglades.

Figure 12 Mean interferometric coherence of the marsh wetlands.

The x-axis shows the six interferometric pairs with a short temporal baseline as no more than 24 days.

If the Sentinel-1 data has a potential to be used for monitoring wetland water levels in the study area, it will be able to guide the conservation and restoration of the habitat of endangered Siberian Cranes. However, unfortunately, the Sentinel-1 C-VV data did not perform well in monitoring the water level changes in the marsh wetlands in the study area, although its coherence was high enough (Figs. 11 and 12). A possible explanation for this is that the specific vegetation characteristics and environmental variables in the study area may have caused insufficient double bounce backscattering. In addition, due to the limited calibration data, we are not sure whether Sentinel-1 data has a good performance in the interferometric pair of July 26, 2016 and August 07, 2016. Our finding is contrary to those from several previous studies (Alexakis, Stavroulaki & Tsanis, 2019; Hong & Wdowinski, 2017; Kundu et al., 2017; Liao & Wdowinski, 2018; Zhang et al., 2018) which have shown that the Sentinel-1 C-band data can be used to monitor wetland water level changes in other regions. Nevertheless, the analysis of the potential use of Sentinel-1 C-VV data for detecting wetland water level changes undertaken here, has extended our knowledge of the applicability of such data. Further studies are needed in the future to discern whether a better accuracy could be obtained in other stopover sites for Siberian Cranes with different vegetation characteristics and environmental variables when using Sentinel-1 InSAR technique for detecting water level changes.

Conclusions

This study utilized 28 repeat-pass Sentinel-1 imagery to comprehensively investigate SAR backscatter and coherence variation for marsh wetlands in China’s Momoge National Nature Reserve, a Ramsar recognized wetland site of international importance. The higher temporal resolution of the Sentinel-1 images allowed us assessing the potential of using InSAR applications for wetland dynamic analysis. Our study showed that coherence was mainly dependent on the temporal baseline, not affected by the perpendicular baseline for Sentinel-1 C-VV data collected over the marsh wetlands. A negative correlation between Sentinel-1 coherence and water depth was found for the marsh wetlands during the early growth stage of vegetation, indicating the role of water depth in determining the coherence during the early growth stage of vegetation. For the late growth stage, marsh vegetation canopy can play a key role in determining the coherence. A nearly linear negative relation between Sentinel-1 C-VV coherence and backscatter were found for the marsh wetlands, suggesting that a high coherence is a good indicator of a low Sentinel-1 backscattering response for the marsh wetlands. We found that although the quality of the interferogram largely depends on the coherence, the stage during the growing season, rather than the coherence, is the primary factor in determining the quality of the interferogram for the marsh wetlands. Our findings demonstrate that Sentinel-1 C-VV data is generally not suitable to be used for monitoring water level changes through InSAR technique in marsh wetlands of the China’s Momoge National Nature Reserve because of its poor accuracy.

Supplemental Information

Supplemental Information 1 Raw data

Click here for additional data file.

Additional Information and Declarations

Competing Interests

Author Contributions

Data Availability

The authors declare there are no competing interests.

Yueqing Chen and Lili Wu conceived and designed the experiments, performed the experiments, analyzed the data, prepared figures and/or tables, authored or reviewed drafts of the paper, and approved the final draft.

Sijia Qiao and Liwen Chen performed the experiments, analyzed the data, prepared figures and/or tables, and approved the final draft.

Guangxin Zhang conceived and designed the experiments, authored or reviewed drafts of the paper, and approved the final draft.

Y. Jun Xu conceived and designed the experiments, analyzed the data, authored or reviewed drafts of the paper, and approved the final draft.

The following information was supplied regarding data availability:

The raw data are available in a Supplementary File.

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
