# Peer review of "Investigating the potential use of Sentinel-1 data for monitoring wetland water level changes in China’s Momoge National Nature Reserve"

_PeerJ, doi:10.7717/peerj.8616_

## Round 0.1 · original submission · Major Revisions

Many thanks for your submission, which I and both reviewers agree is a good paper. In addition to the minor corrections identified, there are two areas needing more substantial improvement:

1. the paper needs to be reworked to bring it up to date, particularly in the literature review regarding use of Sentinel 1 imagery
2. a clearer relevance for readers of PeerJ- an environmental sciences journal- should be established. I suggest that the introduction and relevant parts of the discussion should place a greater focus on the need for and importance of wetland monitoring, leading to the methods presented as a possible solution.

I expect that these changes shouldn't take too long and I look forward to receiving the revised manuscript in due course.

Reviewer 1 ·

Basic reporting

One important point on this paper:
Although the paper is acceptable to publish in journal but I am not sure it fall within the scope of Peerj journal. In my idea it is suitable for a journal with remote sensing scope.

Experimental design

No comment

Validity of the findings

No comment

Additional comments

Although the paper is acceptable to publish in journal but I am not sure it fall within the scope of Peerj journal. In my idea it is suitable for a journal with remote sensing scope. But if editor believes it falls within the scope it is with minor revision accept.

The paper entitled “Investigating the potential use of Sentinel-1 data for monitoring wetland water level changes in China’s Momoge National Nature Reserve” aims to systematically investigate the potential of using Sentinel-1 C-VV data to change detection of water level. The paper is presented in a good structure and it described the method and results clearly. Although with some more discussion it would be complete.


In addition some corrections are required as follow:
Introduction:
Although introduction provides valuable information for readers but it’s too long. Please shorten this section
Line 12-19: Please remove the repeated words same as: Wdowinski et al. (Wdowinski et al. 2004; Wdowinski et al. 2008)
Data
Line 131-134: Remove these lines, the OLI is known by readers.
Method
Line 168: Please remove “The method of selection of training samples was similar to that published in literature”. The refrence “(Jia et al. 2014) “ is enough.
Line 175: although MLC is most widely adopted parametric classification but it is old method, and in the recent papers the methos same as SVM, ANN and … are popular. Please remove lines 174-176.
Line 188-190: The references are old . In addition, 9 references are not required to describe these popular terms. Two references are enough.
Line 247: 2.4.5. Best-fit Analysis : please provide more information about Best-fit Analysis
Figures: Please increase the quality of figures. Some figures are not readable same as figure 11

Reviewer 2 ·

Basic reporting

The article titled "Investigating the potential use of Sentinel-1 data for monitoring wetland water level changes in China’s Momoge National Nature Reserve" is well written, well structured, and the figures and tables are significant. The topic is novel and the use of Sentinel-1 adds significant value to the paper.

However, some things need to be addressed before it can be considered for publication:

The references are old. There is no literature review for 2016-2019. There are a lot of studies conducted in the last 2 years that need to be considered.

Please check the following article:
Monthly Analysis of Wetlands Dynamics Using Remote Sensing Data

There is a similarity with the results, you need to compare it with your results.

Experimental design

The methodological part of the study seems logical and meaningful.

Validity of the findings

The results are well structured and supported with meaningful measurements.

Additional comments

Normally I would give this article a minor revision, but the literature leak is a big disadvantage to the manuscript as many studies have been conducted in the last 2 years with Sentinel-1.

---

## Round 0.2 · accepted · Accept

Thanks for the making the changes to your manuscript, which have helped greatly to ensure clarity and bring it up to date.

Reviewer 1 ·

Basic reporting

no comment

Experimental design

no comment

Validity of the findings

no comment

Additional comments

no comment